# Update on Sentinel Lymph Node Methods and Pathology in Breast Cancer

**DOI:** 10.3390/diagnostics14030252

**Published:** 2024-01-24

**Authors:** Jules Zhang-Yin, Etienne Mauel, Stéphanie Talpe

**Affiliations:** 1Department of Nuclear Medicine, South Luxembourg Clinic, Vivalia, 6700 Arlon, Belgium; 2Department of Surgery, South Luxembourg Clinic, Vivalia, 6700 Arlon, Belgium; etienne.mauel@vivalia.be; 3Department of Pathology, South Luxembourg Clinic, Vivalia, 6700 Arlon, Belgium; stephanie.talpe@vivalia.be

**Keywords:** sentinel lymph node, preoperative detection, intraoperative pathology

## Abstract

Breast cancer stands out as the most commonly diagnosed cancer among women globally. Precise lymph node staging holds critical significance for both predicting outcomes in early-stage disease and formulating effective treatment strategies to control regional disease progression in breast cancer patients. No imaging technique possesses sufficient accuracy to identify lymph node metastases in the early stages (I or II) of primary breast cancer. However, the sentinel node procedure emerges as a valuable approach for identifying metastatic axillary nodes. The sentinel lymph node is the hypothetical first lymph node or group of nodes draining a cancer. In case of established cancerous dissemination, it is postulated that the sentinel lymph nodes are the target organs primarily reached by metastasizing cancer cells from the tumor. The utilization of the sentinel node technique has brought about changes in the assessment of lymph nodes. It involves evaluating the sentinel node during surgery, enabling prompt lymph node dissection when the sentinel node procedure is positive. Additionally, histological ultra-stratification is employed to uncover occult metastases. This review aims to provide an update of this valuable technique, with focus on the practical aspects of the procedure and the different histological protocols of sentinel node evaluation in breast cancer.

## 1. Introduction

The sentinel lymph node (SLN) is the first lymph node (LN) to receive drainage from a tumor. The concept of the SLN dates back to the 1970s. It is based on the notion that metastatic cancer cells spread sequentially through the lymphatic system. The tumor cells pass through an initial LN relay before spreading to the other LNs in the drainage chain. A study of this SLN would reflect the invasion status of the lymphatic territory draining the affected organ. It is the LN most likely to contain metastatic tumor cells. LN dissection is only performed if the SLN is the site of metastasis. The SLN technique was first used in penile cancer [1] and then in cutaneous melanoma [2], before being used in the treatment of breast cancer (BC) in 1994 [3,4].

It is in BC that this technique has seen the greatest development, probably because of its incidence. Indeed, it is the most commonly diagnosed cancer among women globally [5]. It is important to have a precise LN staging for both prognostication and treatment in individuals diagnosed with BC, particularly in early-stage disease, as it significantly impacts the prognosis and aids in regional disease control [6]. Clinical examination, such as palpation, lacks the precision needed for assessing axillary status. Even the modern preoperative imaging techniques, including ultrasound and [18F]-FDG PET/CT, can recognize possibly suspicious LNs; however, they demonstrate limited sensitivity, especially when it comes to detecting micro-metastatic disease [7]. Furthermore, tissue analysis is essential to achieving a more precise assessment of local nodal staging. Nevertheless, axillary lymph node dissection (ALND) frequently leads to a high occurrence of postoperative complications. Furthermore, in early cases of BC, almost 80% of axillary dissections uncover no metastasis, making the surgery mostly ineffective [8].

In the past three decades, there has been a significant shift in the clinical approach to managing the axilla in BC patients. The focus has transitioned towards minimizing surgical interventions to avoid overtreatment. Thus, the SLN biopsy has predominantly supplanted ALND, especially in patients without clinically apparent LN involvement [9]. The SLN technique is really useful for improving the quality of life of patients: it avoids the unnecessary LN dissection, thereby reducing the length of hospitalization, the risk of infection in the axillary fossa, and the risk of lymphedema.

The SLN technique also considerably changed the way pathologists dealt with axillary LNs. Axillary LNs were included in paraffin in their entirety, but examined on a single section (one hematoxylin and eosin (H&E) slide per node). However, as the SLN is more likely than non-SLNs to be metastatic, and as this metastasis may be occult (i.e., very small in size), pathologists intensified their study of the SLN. They carried out recut, serial sections (examinations of almost the entire LN on different levels of sections) and additional studies (immunohistochemistry (IHC) and molecular biology), making it possible to visualize metastases that were too small to be detected on a simple H&E section. This histopathological “ultra-stratification” of the SLN resulted in the detection of increasingly small LN metastases (micro-metastases), or even isolated tumor cells, which could not be visualized by conventional lymphadenectomy, and led to a change in the TNM classification of breast cancers in 2002 [10]. A tumor site measuring between 0.2 and two millimeters or more than 200 cells is defined as a lymph node micro-metastasis (pN1mi) and a tumor site measuring less than 0.2 mm or isolated tumor cells fewer than 200 cells, as pN0 (i+), if detected by IHC, or as pN0 (mol+), if detected by molecular biology [10]. This specification was amended for BC in the eighth edition of the TNM [11].

The aim of this article is to provide an update on this SLN technique in the determination of BC, with a focus on the practical aspects of the procedure within a nuclear medicine department and the procedures for the pathological, anatomical, and cytological examination of the SLN(s) removed during surgery.

## 2. The SLN Concept in BC

### 2.1. Physiopathological Features

Embryologically, the breast develops from ectodermal tissue and extends from the surface of the skin. Consequently, its lymphatic drainage pattern is similar to that of the overlying skin. The mammary gland lies between the superficial (subdermal) and deep (subcutaneous) lymphatic plexuses, which are interconnected by a dense network of lymphatic vessels. Traditionally, the lymphatic vessels surrounding the mammary lobules were thought to drain primarily to the subareolar Sappey plexus, part of the superficial plexus of the skin. However, recent evidence suggests that mammary lymph primarily follows a direct course to the nodal basins, bypassing Sappey’s plexus, which has not been demonstrated in cadavers [12]. It is important to note that the pattern of lymphatic circulation in vivo may differ from that observed post mortem.

Current knowledge suggests that lymphatic drainage from the breast is multidirectional, but predominantly to the ipsilateral axilla. Roughly 3% of lymph drainage from the breast is directed towards nodes located in the internal mammary chain [13], with even smaller amounts directed to other nodes such as periclavicular, paramammary, intercostal, interpectoral, contralateral breast, or abdominal nodes [14].

### 2.2. Methodologies in Nuclear Medicine

#### 2.2.1. Generalities

Lymphoscintigraphy for SLN biopsy is suitable for individuals diagnosed with invasive breast cancer of tumor staging T1 and T2 (tumor size < 5 cm), without indications of axillary or distant metastases. In cases where the axillary status will not impact adjuvant treatment decisions, SLN biopsy may be excluded. Significantly, in elderly patients aged 70 and above diagnosed with early-stage, HER2-negative, hormone receptor-positive BC, the omission of SLN biopsy may be considered.

Typically, the procedure for SLN biopsy involves the interstitial injection of a tracer, preoperative scintigraphic imaging, and the use of an intraoperative gamma probe to locate and surgically remove the identified radioactive LN. Although there is agreement within the medical community on certain fundamental aspects of SLN protocols for BC, this is not the case for all practical and technical components. Ongoing debates persist on various aspects, including the radiotracer’s particle size, the preferred injection pathway, the optimal timing and specific methodology of scintigraphy, as well as the techniques employed for intraoperative detection, and the consideration of surgical removal and analysis of extra-axillary LNs. The choice of a specific radiotracer and technique is also influenced by local availability, regulations, and practice.

From a nuclear medicine perspective, three key parameters determine the optimal tracer delivery technique for radio-guided SLN biopsy: the injected activity, injection volume, and injected site. An additional crucial factor to consider is the time gap between injection and surgery, as it directly influences the necessary dosage of injected radioactivity.

#### 2.2.2. Radiopharmaceuticals

The ideal characteristics of a radiotracer include rapid movement to SLNs along with extended and lasting presence within the nodes. The drainage, dispersion, and elimination of the radiotracer from the injection site through the lymphatic system can exhibit variability, influenced by the individual patient’s pathophysiological characteristics and the size of the particles. Generally, smaller particles are drained and cleared first, while larger particles are drained and cleared later, potentially being retained longer at the injection site.

Efficacy in locating axillary SLNs is not significantly affected by radiotracer particle size [15]. Therefore, the choice of radiotracer should be based on local availability rather than differences in the characteristics of lymphatic mapping for SLN detection. Nevertheless, it is generally agreed upon that a radiocolloid with particles between 100 and 200 nm represents the optimal compromise, striking a balance between rapid lymphatic drainage and optimal retention in SLNs [15].

Radiopharmaceuticals commonly employed for radio-guided SLN biopsy in BC include 99mTc-albumin nano-colloid (particle size: 5–100 nm), 99mTc-sulfur colloid (particle size: 15–5000 nm, typically filtered for particle size range restriction), and 99mTc-antimony trisulfide (particle size: 3–30 nm) [16]. Recent advancements have led to the development of new tracers, with the most current commercially available option being Tilmanocept (Lymphoseek^®^). It is composed of a dextran backbone featuring numerous glucose and mannose residues labeled with diethylene triamine penta-acetic acid (DTPA) for the ^99^mTc.

Tilmanocept offers potential advantages due to its small molecular size (7.1 nm) and the receptor-targeted nature of the mannose moieties in 99mTc-Tilmanocept [17].

These properties contribute to rapid movement from the primary site to the SLN, selective accumulation within the node, prolonged storage, and restricted passage to secondary nodes. Recent reports have shown that these characteristics result in high SLN visualization and detection rates compared to the sulfur colloid tracer [17,18].

#### 2.2.3. Injection Procedures

The use of small volumes with a high specific activity for the optimal detection of SLNs is supported by a large body of literature [19]. Large-volume injections can significantly increase the interstitial pressure at the injection point, potentially modifying the lymphatic drainage pattern and redirecting drainage pathways compared to the baseline conditions of interstitial pressure. There is currently no consensus on the recommended activity for an SLN procedure in patients with BC. In current practice, a total injected activity of 5 to 30 MBq, depending on the time interval between scintigraphy and surgery, is generally considered sufficient for same-day surgery. It has been shown that injections and imaging on the preceding day are practicable by appropriately augmenting the quantity of injected radioactivity, reaching up to 150 MBq [19].

The injection site is categorized into two types: peritumoral injection and peri-areolar injection.

Peritumoral injection was an integral part of the original SLN approach and was first introduced in early studies of SLN techniques [20]. In this approach, two measured quantities are injected on either side of the tumor, and it is widely recognized as the preferred method for accurate detection of SLNs in many medical centers. The recognition of this technique’s excellence is attributed to the strategic injection of the tracer into the lymph vessels near the tumor’s drainage pathways. However, these methods require a thorough review of a patient’s prior imaging and medical records, particularly in the case of non-palpable tumors, and have certain limitations, particularly in patients with non-palpable and/or multifocal tumors [19].

The peri-areolar injection, applicable even in cases of non-palpable tumors, has become popular primarily for its practicality in requiring minimal training and its effectiveness in detecting SLNs in the axilla. It typically involves two to four injections at the edge of the areola, presumably aimed at the Sappey plexus. The reasoning behind this administration method is grounded in the widely embraced concept that lymph flows from the intra-/subdermal space to the subcutaneous plexus, despite recent challenges to this prevailing notion. This approach presents various advantages, such as simplicity, a reduced interval between tracer injection and SLN identification, and heightened uptake of the radiotracer by lymph nodes. These benefits contribute to enhanced rates of nodal identification [12].

Numerous studies indicate that all injection modalities effectively identify axillary SLNs, with satisfactory detection rates documented for every injection method concerning the axilla. In contrast, peritumoral tracer administration is associated with significantly higher detection of extra-axillary SLNs. After employing this technique, lymphoscintigraphy shows that drainage to the internal mammary chain occurs in 20% to 30% of cases. However, this percentage is notably lower (less than 3%) following administration around the areola [21]. As a result, specifically for axillary staging, opting for a peri-areolar injection may be advantageous due to improved and quicker visualization of the axillary SLN. Nevertheless, in cases where the surgical strategy encompasses extra-axillary SLNs, the use of peritumoral injection typically improves their detection. Choosing the most suitable injection method seems to depend on particular clinical indications. For individuals with minor or surface-level tumors located in the upper side area of the breast, injections around the areola might be sufficient. This is particularly relevant when aiming to reduce the requirement for avoidable ALND. Conversely, peritumoral injection should be considered for deep or medial tumors. Especially in cases where precise staging, including SLNs beyond the axilla, is necessary, alternative injection methods may be more appropriate [22]. Furthermore, the integration of two injection techniques in a single patient is increasingly being recommended in selected centers, as it has the potential to enhance SLN detection, reduce non-visualization rates, and offer comprehensive benefits [23].

#### 2.2.4. Imaging Procedures

Lymphatic mapping enables the determination of the quantity of LNs situated along a direct drainage pathway and the precise localization of these SLNs within the anatomical space. It is highly recommended to undergo preoperative imaging due to the variability in the lymphatic drainage of the breast into both axillary and extra-axillary nodes. As a result, preoperative lymphatic mapping holds the promise of improving accuracy, especially for extra-axillary LNs, and reducing morbidity compared to the exclusive use of handheld gamma probes [19]. The use of preoperative imaging also serves as a quality control measure, ensuring the proper application of the tracer, preventing injection or radiopharmaceutical failures, and facilitating the correct management by ensuring injection on the appropriate side (left/right). The decision not to employ preoperative lymphoscintigraphy is typically based on logistical considerations.

Studies have demonstrated that incorporating single-photon emission computed tomography/computed tomography (SPECT/CT) allows for the identification of additional SLNs that may not be visible on planar images in a significant percentage of patients [24]. Fusion SPECT/CT imaging typically provides valuable additional information before surgery to surgeons in the majority of cases, resulting in improved localization, shorter operating times, and increased confidence in the technique [25]. However, only a small subset of BC patients undergoing SLN biopsy appear to benefit from SPECT/CT imaging. Implementing this approach is linked to escalated costs, heightened radiation exposure, and prolonged imaging time. Therefore, it is crucial to establish specific criteria for the utilization of SPECT/CT imaging to ensure that for most patients who would not derive significant benefits from this imaging technique it is avoided due to its undue inconvenience. Given the considerable 60% discrepancy in territory between planar images and SPECT/CT, along with the notably enhanced visualization of SLNs with SPECT/CT, this approach is especially recommended for patients experiencing cancer recurrence on the same side following previous breast surgery or radiotherapy [26] (Figure 1).

#### 2.2.5. Preoperative Detection of SLNs

In the operating room, the surgeon uses a gamma probe to locate the SLN before opening the skin, with the help of lymphatic mapping. The SLN is removed and checked to confirm that it is still radioactive ex vivo and the axillary fossa is also checked to ensure that no activity remains. Then, the SLN is immediately sent to a pathology department for extemporaneous examination.

In around 1–2% of patients, SLNs may go undetected both before and during surgery, making it challenging to ascertain the status of axillary LNs [19]. Factors such as obesity, advanced age, tumor location outside the upper outer quadrant, and lack of SLN visualization on preoperative lymphoscintigraphy can contribute to unsuccessful SLN localization. Fortunately, most patients who do not exhibit SLN visualization in preoperative lymphoscintigraphy will likely have at least one SLN identified during surgery, either through the use of a gamma probe alone or a combination of a gamma probe and blue dye [19].

There is currently no consensus on how to proceed if no SLN is identified during surgery. A potentially promising option in cases of non-visualization is to administer a second dose of tracer, with reserved consideration of SPECT/CT imaging in patients who continue to demonstrate a persistent lack of drainage after re-injection. However, given the increased risk of SLN metastasis in these individuals, established standards of care recommend ALND if intraoperative SLN identification is not achieved [27].

## 3. Pathological Examination of SLNs in BC

### 3.1. Generalities

The past decade has witnessed a transformation in practice-changing trials, thereby impacting the pathological examination of SLNs in BC patients. While the importance of intraoperative SLN diagnosis and the application of advanced protocols for detecting minute metastatic deposits have been subjects of debate in early-stage BC cases, a distinct consideration is imperative when determining the best procedure for SLN assessment in BC patients undergoing neoadjuvant therapy. However, there persists a degree of variability in the practices adopted by pathologists, encompassing the method of intraoperative examination, the strategy for gross processing, and the application of staining levels with H&E as well as cytokeratin IHC [28].

### 3.2. Intraoperative SLN Examination (Extemporaneous Examination)

Extemporaneous examination involves the analysis of SLNs during surgery to detect invasion, enabling complete LN dissection in the same procedure if the SLNs show positivity, thus avoiding the need for a subsequent operation. Various techniques are available for extemporaneous SLN assessment:

Macroscopic examination—This entails palpating the node, and the presence of a whitish indurated area indicates macro-metastasis. Macroscopic examination is used in conjunction with one of the three techniques described below to enhance sensitivity [29].

Extemporaneous cytological examination (Figure 2A–E)—This method includes slicing the LN into two halves, placing each slice on a glass slide, transferring cells (lymphocytes and potential metastatic cells) to the slide, and obtaining an imprint of the slice. After staining, typically with toluidine blue, cytological analysis is conducted to identify metastatic cells (Figure 3). The advantages of this technique include speed, cost-effectiveness, and the preservation of LN tissue for definitive examination.

Extemporaneous histological examination (Figure 2A,B,F,G)—This involves cutting the LN into two halves, freezing one or both slices, and identifying areas of metastatic invasion after staining the frozen section. Material loss is possible but limited. The frozen section can be supplemented by a rapid IHC study using an anti-keratin antibody [30].

Extemporaneous molecular biology examination—This technique employs molecular biology methods to detect the presence of the gene coding for cytokeratin 19 in LN tissue. Two techniques, GeneSearch^®^ (Veridex, Warren, NJ, USA) and one-step nucleic acid (OSNA^®^) amplification (Sysmex, Kobe, Japan), are utilized. For instance, OSNA^®^ involves amplifying mRNA directly from tissue lysates, obtained from half of the SLN, and examining the other half by standard histology. Results, categorized as (++), (+), or (0), are based on the number of gene copies, correlating with the number of tumor cells. The primary drawback is the inability to conduct histological examination on half of the compressed LNs. It is a more expensive and time-consuming technique than the cytological and histological frozen section techniques, with potential limitations in detecting carcinomas not expressing cytokeratin 19 and susceptibility to false positives from benign epithelial inclusions [31,32].

### 3.3. Definitive Anatomopathological Examination

The SLN’s definitive examination is histological. The SLN can be halved along its longest axis and then embedded in paraffin [33]. However, in most instances, creating fine macroscopic slices at two-millimeter intervals ensures the thorough detection of macro-metastases [34]. It is advisable to perform macroscopic slicing along the LN’s longest axis, as this aligns with the afferent lymphatic vessel’s penetration of the node, enhancing the visualization of subcapsular metastases [34].

Following this, the SLN undergoes histological ultra-stratification, involving examination at multiple levels of histological sections and continuous cutting of the paraffin block until it is nearly exhausted. This comprehensive approach ensures that almost all of the LN parenchyma, or at least the majority, is examined under the microscope. Serial H&E-stained sections significantly enhance the detection of occult metastases by 7–10% [35]. However, there is ongoing debate regarding the optimal number of sections to be taken and the spacing between them. Given the conversion rate suggesting that a single slice level is insufficient, SLN ultra-sectioning is deemed essential.

Isolated tumor cells (ITCs), defined as clusters smaller than 200 microns or with fewer than 200 cells, can be identified by spacing sections 150 to 200 microns apart. The combination of serial sections with an IHC study further amplifies the detection of occult metastases in the SLN by 10 to 20% [15,35].

Considerations of the SLN technique in BC.

The SLN technique is a standard in the detection of BC. For many years, the majority of teams performed an extemporaneous examination of the SLN in order to complete the operation with an ALDN at the same time, in the event of extemporaneous positivity of the SLN.

The two techniques, cytology and histology, have fairly similar sensitivity, ranging from 55 to 91% for the former and 57 to 87% for the latter [36,37], but some studies clearly show that histology (88.2% sensitivity) is superior to cytology (47.1%) [38].

In a literature review, freeze sectioning had a sensitivity of 57% to 74% (80% for macro-metastases and 11–27% for micro-metastases) and apposition cytology, a sensitivity of 53% to 91% (81% for macro-metastases, 22% for micro-metastases, and 0% for isolated tumor cells). The false-negative rate varies from 3 to 10% for cytological apposition to 6 to 24% for frozen sections. Specificity is excellent (100%), with the exception of a few cases of false positives with cytology (less than 1%) [39]. In a large series including 2137 patients, the sensitivity of cytology was 53% [40]. Cytological apposition can be performed after macroscopic sections of the lymph node every two to three millimeters, in which case the sensitivity increases to 85% [41].

The combination of careful macroscopic examination of the SLN, cytological staining, and freeze sectioning increases the sensitivity of the extemporaneous examination to 83%. In fact, a study comparing different techniques for extemporaneous examination of the axillary SLN in BC showed a sensitivity of 50% for apposition cytology, 72% for freeze sectioning, 78% for cytology combined with freeze sectioning, and 83% for freeze sectioning combined with rapid extemporaneous IHC (the definitive paraffin section being the gold standard). However, this increase in sensitivity for IHC was mainly due to the detection of micro-metastases, as 100% of macro-metastases were detected by freeze sectioning alone in this study [30].

A meta-analysis, including 12 eligible articles and totaling 2192 patients and 5057 axillary SLNs, compared the OSNA^®^ technique versus histological ultra-stratification [31]. The rate of detection of macro-metastases was equivalent between OSNA^®^ (429) and histology (432). However, 21% of SLNs classified as macro-metastatic with OSNA^®^ were in fact non-macro-metastatic using histology and would not necessarily have led to axillary dissection. Furthermore, in 2% of triple-negative and grade 3 breast carcinomas, the low expression of CK19 led to a negative result with OSNA^®^ and a positive result using histology. OSNA^®^ has a sensitivity of 87%, a specificity of 98%, and a positive predictive value of 79%, and it is not recommended by the authors of this meta-analysis if histological ultra-staging is applied to the SLN [31].

The indications for extemporaneous examination of the axillary SLN were significantly reduced following the publication of the results of the ACOSOG Z0011 study, which showed that there was no benefit from systematic additional axillary dissection in terms of overall survival and disease-free survival for patients with sentinel node metastases [42]. Indeed, the scenarios in which ALND has been performed have become more limited [43,44,45,46]. As reported in a study conducted at a single institution, there was a significant reduction in the utilization of analysis of frozen sections for axillary SLNs, of 69% to 2% [46]. Among patients who meet the criteria of the AMAROS and Z0011 trials, who are clinically node-negative, their SLNs can be straightly submitted for continuous processing, as a tiny proportion is expected to have three or more positive SLNs necessitating completion of ALND [47].

Axillary management in mastectomy patients remains controversial, as they met the exclusion criteria in the Z0011 trial [48] and were not adequately represented in the 23–01 trial [49] (9%) and AMAROS [50] (17%) populations. The decision to utilize intraoperative SLN evaluation may hinge on the clinical scenario and institutional habits [51,52,53]. However, the value of intraoperative SLN evaluation remains evident in patients who underwent neoadjuvant chemotherapy and were clinically node-negative when operated on. In this specific group, ALND remains the standard treatment for any extent of the disease, thereby rendering the intraoperative identification of even minor metastases of clinical importance [54].

When preparing SLNs for intraoperative evaluation, it is essential to diligently identify, dissect, and count each individual lymph node from the collected tissue samples. This may involve the removal of excess fat for better visualization and examination. Ensuring an accurate LN count is particularly crucial to providing the surgeon with comprehensive information, particularly in cases following treatment, with the removal of a minimum of three SLNs being critical to maintaining an acceptable false-negative rate (FNR) [55,56,57]. Insufficient removal may result in the need for ALND. It is vital to confirm the presence of a clipped node, and any localizing markers used, like seeds or reflectors, must be retrieved. Not removing the LN initially proven positive through biopsy could also make ALND necessary [54]. To facilitate histological evaluation, SLNs should undergo sectioning in 2 mm intervals, with those lacking very obvious tumors being completely submitted for analysis.

The intraoperative evaluation of SLNs can be made with various techniques. The most commonly employed one is frozen section analysis, which exhibits an overall sensitivity of 78% [28], with a higher sensitivity for detecting macro-metastasis compared to micro-metastasis (94% versus 40%) [58]. The majority of false-negative results stem from sampling errors (in as many as 94% of cases) [59,60,61,62], where the frozen section slides do not show any visible tumor cells. This underscores the significance of accurately performing serial sectioning of LNs into 2 mm intervals and histological sectioning of frozen tissue, which serves to improve the representation of the tissue surface area on the frozen slides [61,62].

The metastatic disease’s size plays a crucial role in influencing the FNR [37,60,61,62,63,64,65,66,67,68], with reports suggesting an increase of up to eight times in the FNR for micro-metastatic disease versus macro-metastatic disease [63]. Errors in the interpretation of SLN frozen slide analysis are linked with tumor features, including low-grade tumors [59,60,61,69] and lobular histological subtypes [63,64,67,70]. Examining frozen sections of SLNs following neoadjuvant chemotherapy presents a significant challenge, as the cellularity of metastatic deposits is reduced and the alterations caused by the treatment can mask findings. Nevertheless, recent studies suggest that frozen section analysis continues to be a reliable method for evaluating SLNs in the neoadjuvant context, demonstrating a low FNR of only 5.4% [61].

Intra-operative cytology using fingerprints, scrapings, and smears has been advocated for its cost-effectiveness, speed, ease of execution, and superior tissue preservation compared to frozen sectioning. Nevertheless, at a 63% overall sensitivity, touch imprint cytology is less sensitive than frozen sectioning [71]. There is a notable drawback for cytological techniques, as they are unable to accurately determine a metastasis’ size. This limitation may pose challenges in certain clinical scenarios, especially when considering low-volume diseases managed conservatively in patients who do not meet the criteria outlined in the Z0011 trial. Therefore, it is advised that intraoperative cytology should be used more suitably as a supplementary method to frozen section analysis, rather than as an independent approach.

There is a suggestion that SLN intraoperative assessment accuracy could be enhanced by employing rapid IHC and molecular techniques. In comparison to frozen sectioning and touch imprint cytology, rapid cytokeratin IHC is regarded as the least sensitive method for detecting metastases in SLNs [72]. When employed alongside frozen section analysis, the SLN intraoperative assessment’s sensitivity increases, attaining a level of accuracy that is comparable to that of the final pathology [30].

One specific molecular technique, OSNA, employs reverse transcription and loop-mediated isothermal amplification of cytokeratin 19 mRNA to classify SLNs as negative, micro-metastases, or macro-metastases [73]. It demonstrates elevated sensitivity (87%) for identifying macro-metastases; nevertheless, there is a concern that more than 20% of patients who are diagnosed with macro-metastases by OSNA might undergo a reclassification of their disease status as micro-metastases after histopathological examination [31]. A significant criticism of this method is the potential for inconsistent results at crucial nodal staging thresholds. False negatives are possible, particularly with cytokeratin 19-negative tumors, and there is a chance of occasional false positives due to benign inclusions.

### 3.4. Particular Features of SLNs in BC

#### 3.4.1. Invasive Lobular Carcinoma Setting

Special attention is required when dealing with the metastasis of invasive lobular carcinoma (ILC) to lymph nodes (LNs), owing to its distinctive pattern of spread involving non-cohesive tumor cells. There is ongoing debate about the relevance of the Z0011 results for patients with lobular histology, given that they represented only 7% of the study population [74]. In one study with patients who were eligible for the Z0011 trial, those with ILC more frequently exhibited SLN macro-metastases and extra-nodal extension exceeding 2 mm. Nevertheless, the proportion of patients with three or more positive SLNs did not exhibit a significant difference when categorized by histological subtype. This indicates that the decision for ALND is not justified solely on the basis of histology [75].

There exist conflicting data regarding whether lobular histology is linked to a higher risk of involvement in non-SLNs [75,76,77,78,79]. However, the overall node positivity rate between invasive ductal carcinoma and ILC has not been shown to be significantly different in several studies [80,81,82,83,84]. In early BC patients with ILC, isolated tumor cells and micro-metastases are more likely to be found in axillary SLNs.

Regarding the prognostic significance of the detection of these MMs or ITCs. Previous studies have yielded controversial results. New techniques, including IHC, have made it possible to find MMs and ITCs more frequently. Recent studies have tended to support the existence of prognostic significance. Indeed, Luo et al. designed a study to distinguish the prognosis and local treatment recommendations for N1mi BC patients with different numbers of LNMMs involved and demonstrated that for BC patients with an identical T1-2N1miM0 stage, the greater the number of LNMMs, the worse the prognosis (*p*  <  0.001) [85]. Liikanen and colleagues also found that the presence of ITCs in the SNs was an independent predictor of distant recurrence in this cohort of patients with pT1 node-negative early BC [86]. And at last, Merfeld et al. found that T1-T3 N1mi BC patients with grade 3 MMs were at substantial risk for locoregional recurrence [87].

Among patients suffering from classic ILC, interpreting SLNs poses significant challenges due to their scattered unicellular pattern and often blunt cytological features. This difficulty can result in the overlooking of nodal disease, even if they are relatively large [88], when relying solely on routine H&E analysis or frozen sectioning. Among ILC patients, occult metastases in LNs can be identified through cytokeratin IHC in up to 40% of cases [35,89]. Thus, it is advised to routinely employ cytokeratin IHC for evaluating SLNs in cases where the morphology is lobular [90] (Figure 4). This not only helps in detecting metastatic disease, but also helps in understanding its extent.

Because ILC metastasis affects individual cells rather than cohesive clusters, determining the appropriate size of the largest metastatic deposit can be challenging. In the event that there is a scattered pattern of nodal involvement, pathologists are encouraged to use their discretion in assigning the most appropriate N category. Furthermore, it is advisable to document in their report their rationale for the challenges encountered in classification [91].

#### 3.4.2. Extranodal Extension

Extracapsular nodal extension (ENE) is characterized by the extension of metastatic cells through the nodal capsule into the perinodal adipose tissue. The occurrence of ENE in the SLN is reported to be in the range of 24–40% [91]. In the Z0011 trial, gross ENE was a criterion for exclusion. However, the impact of microscopic ENE was not examined. Prior studies have determined that ENE associated with axillary SLN metastasis acts as a crucial indicator of the absence of involvement in sentinel axillary nodal disease, as well as being predictive of disease recurrence and overall mortality [92,93,94,95].

In patients who would have qualified for the Z0011 trial, research has specifically focused on exploring the significance of ENE. Gooch et al. found that 33% of patients with over 2 mm of ENE had four or more positive nodes, compared to only 9% of patients with 2 mm or less of ENE [94]. Another study demonstrated that ENE is correlated with the mean SLN without metastases. However, those with and without ENEs did not differ significantly in terms of recurrence or survival. Some studies have proposed that the existence of microscopic ENE should not be a key factor in recommending ALND [96].

It is advised that not only the presence but also the extent of ENE should be reported [97]. The degree of ENE might predict non-SLN involvement and affect the decision to proceed with ALND. Although there is no established standard for measuring ENE, major guidelines recommend measuring the widest diameter of the invasive front of the ENE, either perpendicular or parallel to the nodal capsule [98]. It is acknowledged that more research is necessary for evidence-based guidelines, and recent studies have been exploring the prognostic significance of varying ENE diameters.

## 4. Conclusions

SLN technique stands as a crucial element of the standard care protocol for BC, ideally integrated into a tailored multidisciplinary approach for each patient. Ongoing efforts to make this procedure universally accessible to BC patients globally are expected to enhance overall patient outcomes.

Despite ongoing innovations in LN mapping, conventional SLN biopsy based on radio guidance remains firmly embedded in evidence-based BC management algorithms. It is a time-tested technique that continues to evolve, informed by scientific data, allowing for increasing personalization of the SLN biopsy in individual cases and broader populations.

The recommended methodologies for the axillary SLNs’ pathological examination in patients suffering from BC were shaped by findings from clinical trials over the past decade. Despite this, the pathological examination of the SN is not always standardized enough, and pathologists should stay informed about these developments and establish laboratory protocols and practices that are grounded in evidence-based approaches.

## Figures and Tables

**Figure 1 diagnostics-14-00252-f001:**
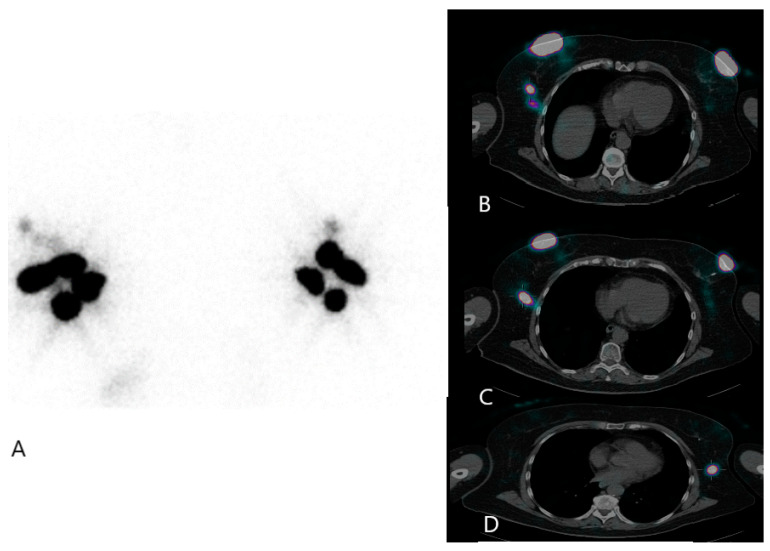
Bilateral breast invasive ductal carcinoma: lymphoscintigraphy after 24 h of injection of 99mTc-albumin nano-colloid: (**A**) planar acquisition; (**B**) SPECT-CT highlighting lymph node 1 of the right side; (**C**) SPECT-CT highlighting lymph node 2 of the right side; (**D**) SPECT-CT highlighting the only lymph node of the left side.

**Figure 2 diagnostics-14-00252-f002:**
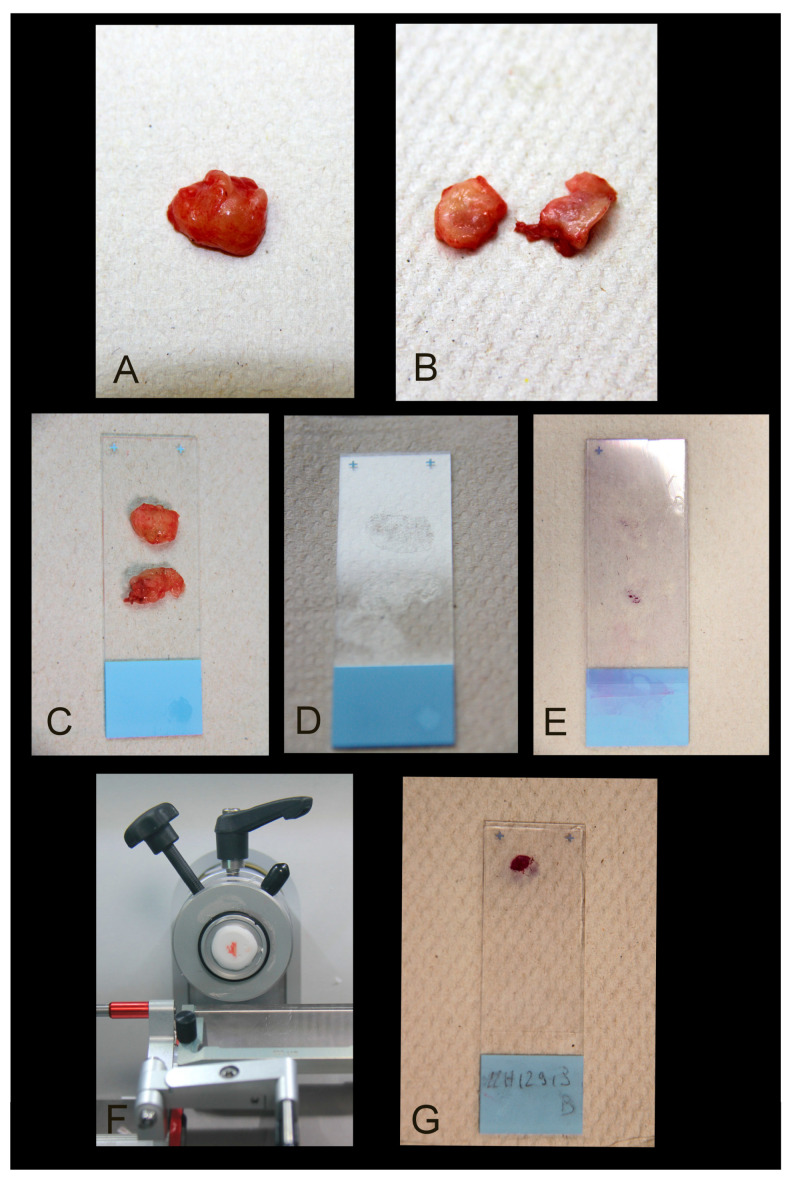
Macroscopic intraoperative SLN process. (**A**) Fresh intraoperative SLN. (**B**) Bivalvular section along the longest axis. (**C**) Slices of LN on a glass slide. (**D**) Unstained transferring cells. (**E**) Stained imprint. (**F**) Frozen slice in a cryostat. (**G**) Stained frozen slide.

**Figure 3 diagnostics-14-00252-f003:**
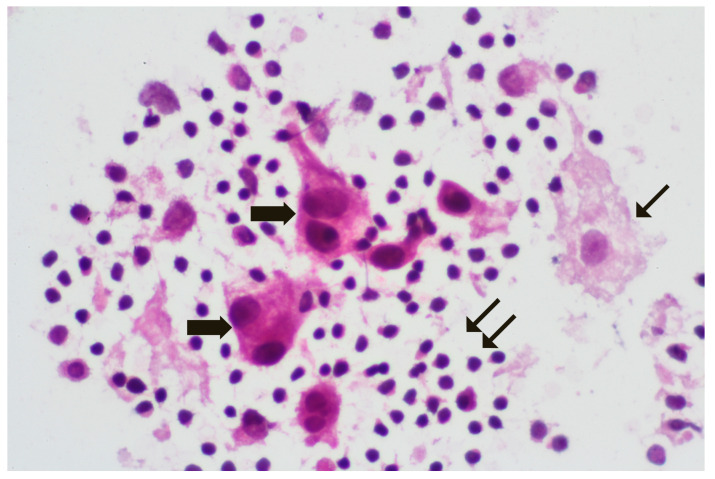
Metastatic sentinel lymph node imprint (rapid H&E), ×100. 🠮 Atypical epithelial cells. ↙ Macrophage. ⇊ Normal LN cells.

**Figure 4 diagnostics-14-00252-f004:**
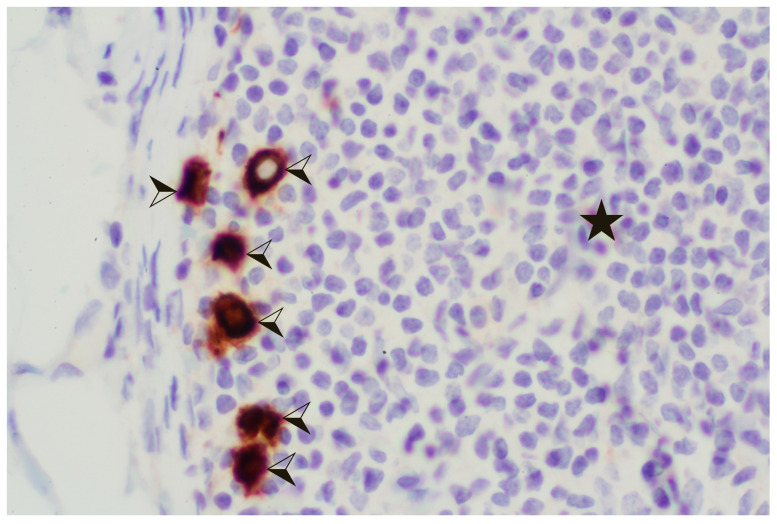
Lymph node with lobular carcinoma, stained with cytokeratin AE1/AE3 in immunohistochemistry, ×100. ★ Normal LN tissue. ⮘ Metastatic stained cells (brown chromogen).

## Data Availability

Not applicable.

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
