# Peer review of "Update on Sentinel Lymph Node Methods and Pathology in Breast Cancer"

_diagnostics, 2024, doi:10.3390/diagnostics14030252_

Round 1
Reviewer 1 Report
Comments and Suggestions for Authors
I believe that the review article named “Update on Sentinel Lymph Node Methods and Pathology in Breast Cancer” submitted by Dr. Zhang-Yin and co-authors, represents an important update on a subject that has been little studied but is essential in determining the prognosis and treatment of breast cancer. It clearly shows the importance of a detailed analysis of the SLN by different techniques including imaging, anatomopathological analysis of the lymph node in its entirety, molecular biology, as well as the limitations of each methodology. The review presents a very realistic picture of the complexity and difficulties of diagnosing macro and micrometastases. I have only a few suggestions to improve the manuscript.
1. Please define DTPA in line 133
2. Page 6, item 3.2. Is it possible to provide some representative images of LN macroscopic processing? I think it would be very helpful for trainees.
3. Figure captions for Fig. 2 and 3 are uninformative. Please improve the legends with more details and indicate specific cells with arrows.
Author Response
We would like to sincerely thank you for your constructive criticism and valuable comments which were of great help in revising this paper. Our responses follow:
1. Please define DTPA in line 133
R: The modification has been made in line 133: DTPA means diethylene triamine penta-acetic acid.
2. Page 6, item 3.2. Is it possible to provide some representative images of LN macroscopic processing? I think it would be very helpful for trainees.
R: A new figure 2 has been added.
3. Figure captions for Fig. 2 and 3 are uninformative. Please improve the legends with more details and indicate specific cells with arrows.
R: Thank you for that valuable suggestion. These figures have been reworked with more detail in the legend and the addition of arrows. They become figures 3 and 4 respectively.
Reviewer 2 Report
Comments and Suggestions for Authors
This is well-constructed and well-written review summarizing the current state of knowledge on the SLN concept in BC. In particular, Authors discuss radiopharmaceuticals, injection and imaging procedures used in nuclear medicine for detection of SLN as well as various aspects of pathological examination of SLN in BC.
I suggest the following corrections:
1. Authors should address the issue of micrometastases (MM) and isolated tumor cells (ITC) in short separate paragraph.
In discussing the TNM classification in the introduction, the Authors note the definition of MM and ITC and the methods that enable their detection. But nowhere in the text do we find information on whether the detection of these MM or ITCs has prognostic significance, and if so, what kind, or whether it affects therapeutic decisions.
Only in the paragraph on ILC we read: “In early BC patients with ILC, isolated tumor cells and micro-metastases are more likely to be found in axillary SLNs, but the clinical significance of this observation remains uncertain [92-94].” However, the Authors cite only three reports without discussing the results.
New techniques, including IHC, have made it possible to find MM and ITCs more frequently. This raises a number of questions. Do they relate to prognosis and treatment recommendations in BC? Is prognosis related to the number of SLNs with MM? Is the prognostic significance of MM and ITCs detected by IHC alone different from those detected by HE?
The Authors should add a short paragraph discussing this issue based on the recent reports published up to 2023 (e.g. Luo S et al. World J Surg Onc 2023, 21:202, DOI: 10.1186/s12957-023-03082-x) and other reports between 2016 and 2023 (references 92-94 are from 2008 and 2016).
2. Figure legends for fig. 2 and 3 are misplaced. Legend for fig. 3 belongs to fig. 2 and vice versa.
3. It should be “two halves” instead of “bivalves” lines 268/269
4. It should be “Pre-operative” instead of “Per-Operatory…” line 222
5. It should be “touch imprint” instead of “touch-print” line 389
6. References should be added after the sentences:
“Thus, it is advised to routinely employ cytokeratin IHC for evaluating SLNs in cases displaying lobular morphology.”
“Previous research has established that ENE linked with axillary SLN metastasis serves as a key predictor of the absence of involvement for sentinel axillary nodal, disease recurrence, and overall mortality.”
“A number of studies have suggested that the presence of microscopic ENE should not be a determining factor in the recommendation of ALND.”
Author Response
We would like to sincerely thank you for your constructive criticism and valuable comments which were of great help in revising this paper. Our responses follow:
1. Thank you for this insightful remark. A new paragraph has been added at L443 with the modification of references: suppression of these
- Truin W et al.
- Mittendorf EA et al.
- Tan LK et al.
And addition of these:
- 92. Luo S, Fu W, Lin J, Zhang J, Song C. Prognosis and local treatment strategies of breast cancer patients with different numbers of micrometastatic lymph nodes. World J Surg Oncol. 2023 Jul 10;21(1):202.
- 93. Liikanen JS, Leidenius MH, Joensuu H, Vironen JH, Meretoja TJ. Prognostic value of isolated tumour cells in sentinel lymph nodes in early-stage breast cancer: a prospective study. Br J Cancer. 2018 May;118(11):1529-1535.
- 94. Merfeld EC, Burr AR, Brickson C, Neuman HB, Anderson BM. De-escalating Locoregional Therapy for Axillary Micrometastases in Breast Cancer: How Much is Too Much? Clin Breast Cancer. 2022 Jun;22(4):336-342.
2. We are sorry for this mistake and have corrected that. These figures also have been reworked with more detail in the legend and the addition of arrows.
3. The modification has been made.
4. The modification has been made.
5. The modification has been made.
6. We have added :
- 98. Patel A, D'Alfonso T, Cheng E, Hoda SA. Sentinel Lymph Nodes in Classic Invasive Lobular Carcinoma of the Breast: Cytokeratin Immunostain Ensures Detection, and Precise Determination of Extent, of Involvement. Am J Surg Pathol. 2017 Nov;41(11):1499-1505.
- 100. Choi AH, Blount S, Perez MN, Chavez de Paz CE, Rodriguez SA, Surrusco M, Garberoglio CA, Lum SS, Senthil M. Size of Extranodal Extension on Sentinel Lymph Node Dissection in the American College of Surgeons Oncology Group Z0011 Trial Era. JAMA Surg. 2015 Dec;150(12):1141-8.
- 101. Shigematsu H, Taguchi K, Koui H, Ohno S. Clinical Significance of Extracapsular Invasion at Sentinel Lymph Nodes in Breast Cancer Patients with Sentinel Lymph Node Involvement. Ann Surg Oncol. 2015 Jul;22(7):2365-71.
- 102. Gooch J, King TA, Eaton A, Dengel L, Stempel M, Corben AD, Morrow M. The extent of extracapsular extension may influence the need for axillary lymph node dissection in patients with T1-T2 breast cancer. Ann Surg Oncol. 2014 Sep;21(9):2897-903.
- 103. Yang X, Ma X, Yang W, Shui R. Clinical significance of extranodal extension in sentinel lymph node positive breast cancer. Sci Rep. 2020 Sep 7;10(1):14684.
104. Barrio AV, Downs-Canner S, Edelweiss M, Van Zee KJ, Cody HS 3rd, Gemignani ML, Pilewskie ML, Plitas G, El-Tamer M, Kirstein L, Capko D, Patil S, Morrow M. Microscopic Extracapsular Extension in Sentinel Lymph Nodes Does Not Mandate Axillary Dissection in Z0011-Eligible Patients. Ann Surg Oncol. 2020 May;27(5):1617-1624.
Reviewer 3 Report
Comments and Suggestions for Authors
The article presented is well written. It covers historical aspects and details of the sentinel lymph node technique procedures, as well as an important discussion about the accuracy of various procedures.
Of changes, I suggest checking the article for a few things. For example, line 95, T1, and T2 are not histological types, but tumor staging. Both the acronym and the OSNA technique are described long after it was first presented.
Author Response
Thank you for this careful reading and identification. The changes have been made:
- L95: we removed the “histological types” and replaced it by “tumor staging”
- The acronym of the OSNA has been added to L278 when it first appeared and the paragraph of L414-416 has been amended.